

# Early detection of student degree-level academic performance using educational data mining

Areej Fatemah Meghji[1], Naeem Ahmed Mahoto[1], Yousef Asiri[2], Hani Alshahrani[2], Adel Sulaiman[2] and Asadullah Shaikh[3]

[1] Department of Software Engineering, Mehran University of Engineering and Technology Jamshoro, Hyderabad, Jamshoro, Pakistan
[2] Department of Computer Science, College of Computer Science and Information Systems, Najran University, Najran, Saudi Arabia
[3] Department of Information Systems, College of Computer Science and Information Systems, Najran University, Najran, Saudi Arabia

## ABSTRACT

Higher educational institutes generate massive amounts of student data. This data needs to be explored in depth to better understand various facets of student learning behavior. The educational data mining approach has given provisions to extract useful and non-trivial knowledge from large collections of student data. Using the educational data mining method of classification, this research analyzes data of 291 university students in an attempt to predict student performance at the end of a 4-year degree program. A student segmentation framework has also been proposed to identify students at various levels of academic performance. Coupled with the prediction model, the proposed segmentation framework provides a useful mechanism for devising pedagogical policies to increase the quality of education by mitigating academic failure and encouraging higher performance. The experimental results indicate the effectiveness of the proposed framework and the applicability of classifying students into multiple performance levels using a small subset of courses being taught in the initial two years of the 4-year degree program.

# INTRODUCTION

For centuries, the method of educating a large set of students has revolved around instructions being passed to them in a classroom setting (*Romero & Ventura, 2013*). An instructor delivers lectures and gives tasks; a student attempts to solve these tasks to the best of his/her ability. By monitoring student class behavior, observing their engagement patterns, and checking their task solutions, the instructor can better assess how well a student has grasped concepts. These observations or feedback help instructors revise and modify course contents and the method of lecture delivery. This feedback is an essential component of higher education systems (*Bransford, Brown & Cocking, 1999*). Sadly, an increase in the number of students in a class makes it difficult for the instructor to obtain

Corresponding author
Yousef Asiri, yasiri@nu.edu.sa

and record this feedback from each student. The absence of this traditional feedback channel necessitates the exploration of other sources of available data that may aid higher educational institutes create additional feedback loops.

Higher educational institutes collect and store vast amounts of student data (*Baek & Doleck, 2022*; *Khan & Ghosh, 2021*). This data includes student demographics, test scores, course assessments, and so on. In recent years, instead of simply storing this data in filing cabinets, an immense amount of research has been conducted on exploring this data to better understand various facets of student learning and behavior. The field of Educational Data Mining (EDM) is an evolving area of research that gained momentum in 2008 (*Khan & Ghosh, 2021*; *Baker, 2014*). To find meaningful patterns and hidden insights in the data emerging from the sector of education, EDM builds on techniques from data mining, machine learning, and statistics to analyze this data (*Viberg et al., 2018*). EDM aims to extract knowledge from educational data and use it for improved feedback and decision-making (*Berland, Baker & Blikstein, 2014*). A unique feature of educational data is the internal hierarchy and correlation amongst data at different levels. Taking this into consideration, EDM approaches explicitly exploit the non-independence and multi-level hierarchy in educational data to predict an overall pattern (*Romero & Ventura, 2020*). There are five key approaches or research areas in EDM: prediction, relationship mining, clustering, discovery within models, and distillation of data for human judgment (*Peterson, Baker & McGaw, 2010*).

Classification is a popular approach within prediction (*Khan & Ghosh, 2021*; *Viberg et al., 2018*). In classification, educational data is fed to an algorithm specifically designed to infer or predict the value of an attribute (class label) based on the patterns or relationships discovered within certain other attributes (predictor variables). Classification has been applied at various levels of granularity to address an ever-increasing set of problems within the educational domain such as inferring a student's emotional state (*Dmello et al., 2008*), predicting student drop-outs (*Agrusti, Bonavolontà & Mezzini, 2019*; *Márquez-Vera et al., 2016*; *Delen, 2010*), developing recommender systems (*Mimis et al., 2019*; *Erdt, Fernandez & Rensing, 2015*), predicting student retention (*Shafiq et al., 2022*), examining the use of learning materials uploaded in an e-learning platform (*Valsamidis et al., 2011*), and to identify patterns associated with student success in e-learning platforms (*Sánchez et al., 2023*). A key application area has been predicting student academic outcomes (*Xiao, Ji & Hu, 2022*; *Nahar et al., 2021*; *Viberg et al., 2018*; *Romero & Ventura, 2020*; *Fernandes et al., 2019*). Research in this area has been carried out to predict student success in a course, their grades in a semester, and, to a smaller extent, their success in terms of exam verdict or grades at the end of a degree (*Asad, Arooj & Rehman, 2022*; *Romero & Ventura, 2013*; *Berland, Baker & Blikstein, 2014*; *Nghe, Janecek & Haddawy, 2007*; *Asif et al., 2017*).

## Goal of the research

By analyzing the most basic student data collected by a higher educational institute, this research aims to devise a classification model that predicts student end-of-degree performance at an early stage during the course of the degree. The goal is to not only predict student performance in terms of academic achievement but also discover courses that

impact this performance. This has been done to provide instructors and policy-makers the feedback needed to meet their objective of creating a student-centric learning environment. The predictions made by the model have also been used to devise a segmentation framework that can effectively classify students into learner categories and further help in designing a pragmatic pedagogical policy.

### Research Questions

In light of the goal of the research, the work presented in this article attempts to answer the following questions:

- Is the generation of a predictive model for early detection of student end-of-degree performance possible using the most basic and readily available learning data collected by higher educational institutes?
- Can courses that strongly influence the final prediction of student end-of-degree performance be ascertained to provide intervention?
- Can a segmentation framework be devised to help design a pragmatic pedagogical policy?

The rest of this article is organized as follows: A review of the related literature has been presented in 'Related Work' followed by 'Classification' which outlines the process of classification, the working mechanism of some popular classifiers used in this article, and the metrices used to evaluate the performance of a classifier. 'Research Methodology' explores the experimental setup of this research followed by the Experimental Results and Discussion. Finally, a conclusion and suggestions for future work have been presented in 'Conclusion and Future Work'.

## RELATED WORK

Higher educational institutes constantly strive to provide an environment that fosters student-centric learning (*Romero & Ventura, 2020*). The proper analysis of data emerging from the sector of higher education has the potential to manifest results that can not only help enhance student performance but also elevate teaching effectiveness. EDM is being increasingly used to improve educational outcomes. In particular, researchers have focused on developing classification models to predict student performance (*Baek & Doleck, 2022*; *Xiao, Ji & Hu, 2022*).

*Nghe, Janecek & Haddawy (2007)* investigated students' undergraduate and postgraduate academic performance at two universities. For Can Tho university in Vietnam, a total of 20,492 undergraduate student records between the years 1995 to 2002 have been explored. The attributes of gender, English language skill, age, family job, and CGPA in the second year of study have been used to predict GPA at the end of the third year of education. Decision trees and Bayesian classifiers have been used to classify student performance. Experiments have been conducted to classify students into four GPA-based classes: fail, fair, good, and very good; three classes: fail, good, and very good; and two classes: fail and pass. The decision tree outperformed in all the experiments. It was observed that accuracy of the classification model increased when the number of class labels was decreased;

classifier performance for four classes was 72.95% which improved to 86.47% when made to predict three classes. The performance further improved to to 94.03% for prediction of two class labels. For the Asian Institute of Technology in Thailand, 936 student records were explored between 2003 and 2005. The attributes of university entrance GPA, proficiency in English, and gender have been used to predict GPA at the end of the first year of the master's program. Here, too, the decision tree outperformed with an accuracy of 70.62% for four classes, 74.36% for three classes, and 92.74% for two classes.

*Miguéis et al. (2018)* explored data of 2459 students belonging to an engineering and technology school of a European public research university between the years 2003 to Student data available after the first year of a degree program has been used to predict degree-level student academic performance. The data for this research included socio-demographic features, social-economic features, high school background, and data of the first year of the degree. Several classification algorithms have been explored, including Naïve Bayes (NB), Sequential Minimal Optimization (SMO), decision trees, and Random Forests (RF). The classification model based on RF exhibited an accuracy of 96.1%.

*Kabakchieva (2013)* analyzed data of 10330 students across 20 attributes between the years 2007 to 2009 in a Bulgarian educational institute. After an initial exploration of data, 6 attributes have been removed, and the study has been conducted using student attributes that, among others, included gender, previous education, score in the university entrance exam, and current semester score. Student performance has been classified into five classes (excellent, very good, good, average, or bad) using the decision tree, NB, K-nearest neighbor, and rule-based classifiers. Although the decision tree-based J48 outperformed, all the classifiers achieved an accuracy of less than 70%. The university admission score was discovered as the most influencing attribute towards the final class prediction.

*Aman et al. (2019)* analyzed data of 1,021 students pertaining to academic, demographic, and socio-economic attributes between the years 2014 to 2017. To ascertain the relevance of the considered category of attributes, experiments were performed using only academic and combinations of academic, demographic, and socio-economic attributes. Some attributes considered in this research include gender, division obtained in previous studies, literacy rate, study mode, and the index of poverty of student residential areas. The best results were found using all attributes.

In contrast to the studies discussed thus far, a significant decrease in the dataset size can be observed in the remaining studies. *Nahar et al. (2021)* have predicted student performance by experimenting on data of 80 students from the department of CSE, Notre Dame University Bangladesh. Student performance has been classified into three categories of good, bad, and medium, based on data from student mark sheets and a behavior survey. Experiments have been conducted using decision trees, NB, RF, and techniques of bagging and boosting. The accuracy of their experiments ranged between 64% to 75% on the test data.

*Zimmermann et al. (2015)* explored data of 171 students belonging to the Bachelor and Master programs in Computer Science at ETH Zurich, Switzerland. The research analyzed how efficiently student undergraduate performance could indicate student graduate-level performance. Using linear regression in conjunction with variable selection strategies,

this research showed that 54% of the variance in graduate-level performance could be explained by undergraduate-level performance. The grade point average of the third year was highlighted as the most significant indicator of overall student performance.

*Asad, Arooj & Rehman (2022)* used the attributes sessional marks and internal marks obtained by different sets of students undertaking five different courses to predict if a student will be safe or at risk of failure in the course. Combining the data across the groups, a total of 176 student records in a bachelor degree program have been analyzed in the research. Experiments have been conducted using decision trees, NB, RF, SMO, and Linear Regression. The accuracy of their experiments ranged between 88% to 95%. One concern in the used dataset is the imbalance in the class labels which may have caused biased results.

*Nieto, García-Díaz & Montenegro (2019)* explored data from students belonging to a public sector engineering university in Colombia. A total of 19 attributes comprising of student academic and certain derived variables (1st, 2nd, and 3rd quartile of grades) have been explored. Classification approaches of RF, decision trees, and SMO have been used with a varying set of feature-selected attributes. The classification model based on SMO achieved the highest accuracy of 84.43% using all 19 attributes.

*Asif et al. (2017)* explored the data of 210 students of a public sector university in Pakistan to predict their degree-level performance. This research analyzed the marks obtained by students in various subjects during the first two years of the university degree. Each subject has been treated as an indicator of the final performance prediction. The findings of the research indicate that student performance at the degree level could be successfully predicted by solely using academic marks. Although the NB classifier exhibited the best results with an accuracy of 83.65%, it was established that all classification models are not human interpretable; a model based on the NB classifier could not be used to visualize the generated model. The model based on the decision tree was used to derive subjects that influenced the degree level performance. The decision tree classifier exhibited an accuracy of 69.23%.

In light of the discussed papers, it is evident that various sets of student learning and descriptive attributes have been used to predict student end-of-degree performance with varying degrees of success. Researchers have explored personal features such as age, gender, marital status, parents' education level and job, as well as student learning data such as marks/grade obtained in high school, marks obtained in university entrance exam, and academic marks across various subjects. Some studies have made use of only academic marks, while others have used either a combination of academic and derived or academic and demographic attributes. Based on the reviewed studies it was observed that classifier accuracy is strongly influenced by the number of class labels being predicted; a greater aggregation of academic performance leads to a higher classifier accuracy (*Asif et al., 2017*; *Nghe, Janecek & Haddawy, 2007*). Another observation was that most studies have focused solely on predicting student performance and not on finding the factors/features that increase or decrease this performance (*Nahar et al., 2021*; *Aman et al., 2019*; *Nieto, García-Díaz & Montenegro, 2019*; *Kabakchieva, 2013*; *Nghe, Janecek & Haddawy, 2007*). The resultant model needs to be interpreted in order to provide feedback necessary for academic improvement (*Xiao, Ji & Hu, 2022*). Based on the explored literature, it

is also apparent that there is no 'best' classification algorithm; different classifiers have outperformed each other in the discussed papers based on the nature of the examined data. A trend that can be seen is that experiments have mostly been conducted using decision trees, NB, RF, and SMO. It remains to be seen, however, if the performance of the final classification model is significantly influenced by varying the number of class labels, using feature selection, and using academic attributes in conjunction with derived and demographic attributes.

## CLASSIFICATION

Classification is a popular approach of prediction which, after learning from a set of data, constructs a model that can be used to predict a designated class label for new and, as yet, unseen data (*Mohammed, Khan & Bashier, 2016*). This process can be broken down into two stages of operation. The first stage is termed the training or learning phase, where labeled educational data are fed to a classification algorithm(classifier) (*Romero & Ventura, 2013*). The classifier examines and analyzes this data and generates a classification model *Quinlan (1993)*. The generated classification model represents the pattern or logic of how the provided data is categorized into one or more class labels. Thus, classification can be regarded as the task of approximating a mapping function $f$ from certain input variables $x$ to discrete output variables $y$ or $y = f(x)$. An important consideration during this stage is ensuring that the dataset used to train the model has a balanced representation of the class labels (*Miguéis et al., 2018*). If the sample used to train the model has a biased or skewed distribution towards the classes, the resultant model might have poor predictive performance, especially towards the minority class (*Hassan et al., 2021*).

Classifiers can be broadly categorized based on their internal mechanism of generating a classification model (*Han, Pei & Kamber, 2011*). Some popular classifiers include decision trees, rule-induction, probabilistic, support vector machines, and memory-based classifiers (*Khan & Ghosh, 2021*).

Decision tree classifiers are so named as the model generated by them resembles a flow chart or tree structure (*Baker & Inventado, 2014*). Every internal node in the tree represents a conditional test. Each branch represents the outcome of the test. Starting from a root node, the tree branches out into internal nodes and branches that finally conclude at some leaf node. The leaf node represents the class label. The root node represents the most significant attribute of the dataset and can be determined using various approaches, including entropy, information gain, and GINI index (*Mohammed, Khan & Bashier, 2016*). J48 is a popular classifier in this category. The RF classifier builds on the concept of decision trees. Instead of generating a single decision tree, the RF generates a forest of decision trees. A class label is established by taking into consideration the output of all the generated trees (*Asif et al., 2017*). A key attraction of decision tree-based classifiers is their simplicity and the fact that the resultant model can be deciphered. The visual representation of the tree can be used to identify attributes that most strongly influence the final prediction of a class label as well as to understand the exact combination of the attributes and their precise configurations that lead to a particular class label (*Viberg et al., 2018*; *Quinlan, 1993*).

Another set of classifiers that generate understandable models are rule-based. If-Then conditions are utilized to generate the target function based on the training data (*Han, Pei & Kamber, 2011*). JRip is a popular rule-based classifier that specifically handles overfitting while learning through reduced error pruning. The NB is a probabilistic classifier that works on the Bayes Theorem. This classifier is quick and resistant to overfitting (*Mohammed, Khan & Bashier, 2016*). SMO is another popular classifier that iteratively trains a support vector machine. It is used to solve optimization problems by incrementally dividing problems into smaller sub-problems (*Han, Pei & Kamber, 2011*).

Unlike classifiers that learn from the training set and then discard it once a mapping function or model of their understanding has been generated, memory-based classifiers store the entire training set. To classify new data items, these classifiers compare the test data with the entire stored training set at run-time (*Mohammed, Khan & Bashier, 2016*) For this reason, these classifiers termed instance-based or lazy. These classifiers are computationally expensive, requiring considerable storage space, especially if the training set is large. However, these classifiers do not make assumptions on the training data and thus are adaptable to problems where the learned assumptions may fail. KStar is a popular memory-based classifier.

The second stage of the classification process is the test phase (*Khan & Ghosh, 2021*). Once the mapping function has been approximated, it is used to predict the class label of new data that the model has not been trained on. Labels for the test data are known yet kept hidden to evaluate the performance of the model. An important consideration while building a classification model is how well the model learns the target function from the training data and how accurately the model generalizes to new data (*Xiao, Ji & Hu, 2022*; *Romero & Ventura, 2020*).

The output of a classification model may be one of the four possibilities: true positive (TP), false positive (FP), true negative (TN), or false negative(FN) (*Zeng, 2020*). Consider a scenario where the data has been categorized into two classes: P and N. A TP is a correct prediction made by the model for class P. Similarly, a TN is a correct prediction made for class N. FP and FN are incorrect predictions. An FP means an incorrect prediction for class P; data that should have been classified into class N has been incorrectly labeled as belonging to class P.

A confusion matrix is often employed for evaluating the performance of a classification model (*Bucos & Drăgulescu, 2018*). Table 1 provides the structure of a confusion matrix for a binary classifier.

Accuracy, precision, recall, and $F_1$ score are some evaluation measures computed using the confusion matrix. Accuracy is a measure of correctness. It is used to evaluate how often the predictions made by a classifier are correct (*Nieto, García-Díaz & Montenegro, 2019*; *Farsi, 2021*). Accuracy can be measured using the formula:

$$Accuracy = \frac{TP + TN}{TP + FP + TN + FN} \tag{1}$$

The recall is the ability of a classification model to find all relevant cases(points of interest) in the provided data. It measures how many instances of interest were predicted

**Table 1  Binary confusion matrix.**

|  | Predicted: P | Predicted: N |
|---|---|---|
| **Actual: P** | TP | FN |
| **Actual: N** | FP | TN |

correctly out of all the instances of interest (*Farsi, 2021*).

$$Recall = \frac{TP}{TP + FN} \tag{2}$$

Precision is used to measure the fraction of instances the classification model considers relevant that actually are relevant (*Aman et al., 2019*; *Hassan et al., 2021*). This metric is used to quantify the correct positive predictions. In other words, it is the ratio of correct positive predictions to all positive predictions made by the model.

$$Precision = \frac{TP}{TP + FP} \tag{3}$$

The $F_1$ score or F-value is the harmonic mean of precision and recall (*Khan & Ghosh, 2021*; *Farsi, 2021*). As it takes into account both FP and FN, it performs well on balanced and imbalanced datasets (*Hassan et al., 2021*). $F_1$ score is measured as:

$$F_1 Score = 2 \times \frac{Precision \times Recall}{Precision + Recall} \tag{4}$$

The weighted $F_1$ score is an average of F-values across all class labels, weighted based on the class distribution (class size). Apart from these evaluation metrics, Kappa is also commonly used to evaluate the performance of a classification model (*Peterson, Baker & McGaw, 2010*; *Fleiss, 1971*). Kappa works under the assumption that a correct prediction could have been made simply by chance. This assumption makes Kappa a useful measure for evaluating the performance of classifiers trained on balanced as well as imbalanced data. Kappa can have a value between 0-1. Similar to accuracy, a higher Kappa value is better; a value above 0.3 signifies that the output of the classifier is not based on chance (*Asif et al., 2017*).

# RESEARCH METHODOLOGY

Figure 1 outlines the research methodology followed in this article. An explanation of each step is provided in the subsequent sections.

## Data collection

The first step of this research was the collection of student data. The current research explores data of students enrolled in the Bachelor of Engineering degree program at the Department of Software Engineering, Mehran University of Engineering and Technology, Pakistan. Data from 291 students belonging to three consecutive batches (13SW: 2013-2016, 14SW: 2014-2017, and 15SW: 2015-2019) has been collected. The data was collected from two main sources: institutional records and individual student files. The Institutional Review Board approved this study with reference number 136.43 on 27-2-2020. The

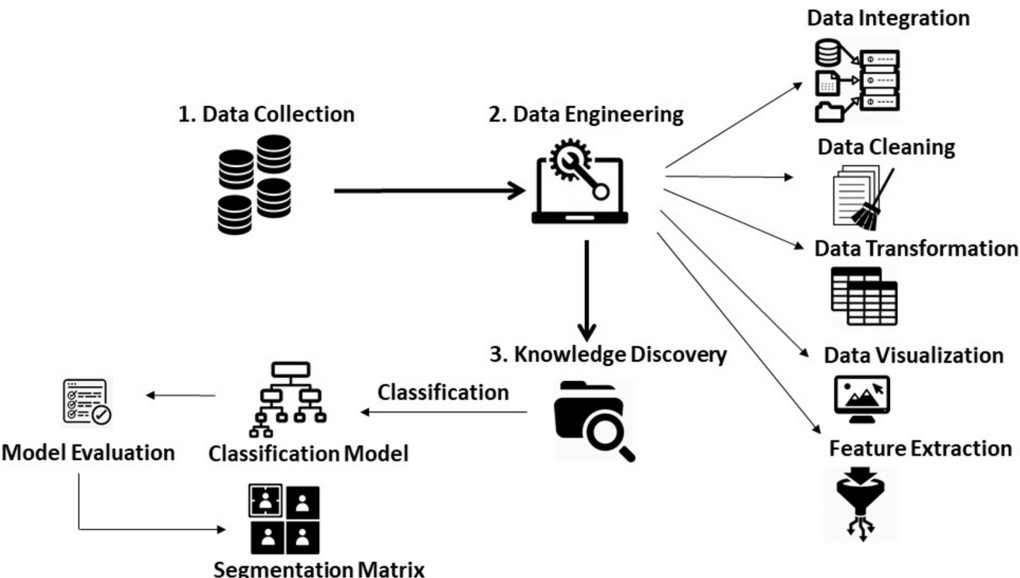

**Figure 1** Research methodology.

institutional records comprised of marks obtained by the student in each subject during the course of the bachelors degree. A total of 28 subjects (theory and practical treated as separate heads) are taught by the end of the 2nd year of education in the Department of Software Engineering. As this research attempts to predict student end-of-degree performance based on student academic achievement in the initial two years of the degree program, every subject has been treated as a feature for the prediction of student end-of-degree performance.

Studies by *Miguéis et al. (2018)* and *Asif et al. (2017)* suggest academic background prior to the enrolment into the university may influence student performance at the university level, thus, the marks obtained in the university admission test, Higher Secondary Certificate (HSC) exams, and Secondary School Certificate (SSC) exams have also been collected through student files maintained by the department. This data was maintained manually and had to be extracted and computerized so that it could be used in this research. The examined literature presents some conflict over the significance and level of influence of the demographic attribute of gender on overall student academic performance (*Khan & Ghosh, 2021*). To examine the influence of gender on the prediction of student performance, this attribute has also been considered in this research.

Building on the premise that derived attributes can play a significant role towards student performance prediction (*Nieto, García-Díaz & Montenegro, 2019*), two derived attributes: 1st year accumulative score and 2nd year accumulative score have been computed; bringing the total number of attributes to 34. A description of some attributes used in this research has been presented in Table 2. The complete list of attributes used in this research, along with their description, has been provided in the Appendix. The data used in this study has also been attached as an additional file named DegreeData_Classification.csv. Although

**Table 2 Attributes in the dataset.**

| Attribute | Description | Type | Value |
|---|---|---|---|
| SSC | SSC Exam Marks | Academic | 0–850 |
| HSC | HSC Exam Marks | Academic | 0–1100 |
| Ad_Test | University Admission Test Marks | Academic | 0–100 |
| ENG11 | Functional English | Academic | 0–100 |
| MTH108 | Applied Calculus | Academic | 0–100 |
| SW111 | Computer Programming | Academic | 0–100 |
| SW111_Pr | Computer Programming Practical | Academic | 0–50 |
| Score_First | 1st Year Accumulated Score | Derived | 0–10 |
| Score_Second | 2nd Year Accumulated Score | Derived | 0–20 |
| Gender | Student Gender | Demographic | M-F |

the student identities have been anonymized by substituting student ids with unique identifiers, this data is not meant for publication as it may be considered sensitive for the university and the students.

## Data engineering
### *Data integration, cleaning and transformation*
After obtaining data of all the attributes considered in this study, the data of all the three batches was integrated into a single dataset. The data was then analyzed to ensure it did not contain missing or erroneous entries. As no missing or null values were uncovered, the data did not require further scrutiny.

As per the policy of Mehran University of Engineering and Technology (set in accordance to the Higher Education Commission of Pakistan), the final percentage of a student in a bachelor degree is computed by the following formula:

$$Final\% = 0.1 \times 1^{st} year\% + 0.2 \times 2^{nd} year\% + 0.3 \times 3^{rd} year\% + 0.4 \times 4^{th} year\%. \qquad (5)$$

The final percentage at the end of the degree is calculated by summing 10% of the percentage obtained in the 1st year, 20% of the percentage obtained in the 2nd year, 30% of the percentage obtained in the 3rd year, and 40% of the percentage obtained in the final year of the degree. Per the marks obtained in the 1st year of the degree program, the percentage of each student at the end of the 1st year has been computed. A similar practice was followed for the 2nd year percentage. Using these values, the accumulated scores or 10% of the 1st year percentage and 20% of the 2nd year percentage have been computed. The computed values of accumulated scores have been treated as derived attributes in this research.

As this research measures academic success in terms of the total percentage obtained at the end of the degree, experiments have been conducted under two settings. First, four classes have been established based on student academic success:
1. Class A: High-Achievers (> =85%)
2. Class B: Above-Average (75%–84%)
3. Class C: Average-Achievers (65%–74%)

**Table 3  Class distribution for the considered batches.**

| Student details | 15SW | 14SW | 13SW | Total |
|---|---|---|---|---|
| Students in Class A | 10 | 16 | 22 | **48** |
| Students in Class B | 40 | 27 | 31 | **98** |
| Students in Class C | 37 | 28 | 22 | **87** |
| Students in Class D | 24 | 15 | 19 | **58** |
| Students in Class SP | 50 | 43 | 53 | **146** |
| Students in Class NI | 61 | 43 | 41 | **145** |
| **Total students** | **111** | **86** | **94** | **291** |

4.  Class D: Under-Achievers ($<65\%$)

Second, two classes have been established based on student academic success:

1.  Class SP: Satisfactory Performance ($>=75\%$)
2.  Class NI: Need Improvement ($<=74\%$)

*Data visualization*  Classifiers are vulnerable when trained on imbalanced class labels (*Hassan et al., 2021*); with imbalanced labels resulting in classification models that provide unreliable and biased predictions. Before proceeding with the experiment, it is important to ensure that each label is well-balanced. The class distribution details provided in Table 3 help analyze the class labels' distribution and ensure the results' authenticity for the next step.

Taking a closer look at the figures provided in Table 3, the class labels for both sets of experiments have a balanced distribution. Classes A and D have slightly lesser representation than classes B and C but the values are within the acceptable percentage (*Khan & Ghosh, 2021*; *Asif et al., 2017*). The classes SP and NI for the second experiment are equally represented.

*Feature selection*  Although the amount of data used to train a classifier has great influence on the effectiveness of the generated model, the size of the data alone does not ensure the accuracy and quality of the generated model (*Asif et al., 2017*). The number of attributes (dimensions / features) being explored, the level of influence these attributes have on the prediction of the class label, and the removal of attributes that inversely affect the prediction of the class label can greatly improve the quality of the generated model (*Matharaarachchi, Domaratzki & Muthukumarana, 2022*). Thus, an important step before knowledge discovery is ensuring the use of optimal attributes for the classifier (*Farsi, 2021*).

CfsSubsetEval is a correlation-based feature evaluator in Weka (*Witten & Frank, 2002*; *Eibe, Hall & Witten, 2016*) which utilizes Pearson's correlation($r$) to determine attributes that strongly influence the prediction of the class label (*Hall, 1998*). As this research uses a large number of attributes, feature selection using CfsSubsetEval has been explored to find the most significant attributes (see Table 4).

Another unique aspect of this research is that feature selection has been applied on the collected data in three stages. First, feature selection has been applied on all the academic attributes only; the demographic attribute of gender and the derived attributes have not

**Table 4 Feature selected attributes.**

| SNo | Experiment - 4 classes | | | Experiment - 2 classes | | |
|---|---|---|---|---|---|---|
| | AC | AC+DR | AC+DR+DM | AC | AC+DR | AC+DR+DM |
| 1 | ES121 | SW125 | Gender | SSC | SSC | Gender |
| 2 | MTH112 | SW215 | SW125 | SW111_Pr | EL101_Pr | SSC |
| 3 | SW121 | SW214 | SW215 | ES121 | SW121_Pr | EL101_Pr |
| 4 | SW122 | SW224 | SW214 | SW122 | SW125 | SW121_Pr |
| 5 | SW125 | SW223 | SW224 | SW121 | MTH212 | SW125 |
| 6 | MTH212 | SW221 | SW223 | SW121_Pr | SW215 | MTH212 |
| 7 | SW211 | SW221_Pr | SW221 | SW125 | SW214 | SW215 |
| 8 | SW214 | SW212 | SW221_Pr | MTH212 | SW211 | SW214 |
| 9 | SW215 | SW222 | SW212 | SW215 | SW223 | SW211 |
| 10 | SW224 | SW222_Pr | SW222 | SW214 | SW221 | SW223 |
| 11 | MTH217 | Score_First | SW222_Pr | SW224 | SW221_Pr | SW221 |
| 12 | SW223 | Score_Second | Score_First | SW211 | SW212 | SW221_Pr |
| 13 | SW221 | | Score_Second | SW223 | SW222 | SW212 |
| 14 | SW221_Pr | | | SW221 | Score_Second | SW222 |
| 15 | SW212 | | | SW221_Pr | | Score_Second |
| 16 | SW222 | | | SW212 | | |
| 17 | SW222_Pr | | | SW221_Pr | | |
| 18 | | | | SW222 | | |
| 19 | | | | SW222_Pr | | |

**Notes.**

AC, Academic; DR, Derived; DM, Demographic.

been used. Second, derived attributes have been added to the academic attributes, and feature selection has been applied to the combination. Finally, all the academic, derived, and demographic attributes have been used. This has been done to better analyze the effect of the features on the final prediction.

## Knowledge discovery

The following steps have been followed for knowledge discovery in this article:

- Following the reviewed literature (*Nghe, Janecek & Haddawy, 2007*; *Miguéis et al., 2018*; *Kabakchieva, 2013*; *Zimmermann et al., 2015*; *Nieto, García-Díaz & Montenegro, 2019*; *Asif et al., 2017*; *Aman et al., 2019*; *Asad, Arooj & Rehman, 2022*), six popular classification algorithms: NB, J48, JRip, RF, SMO, and KStar, have been used to predict student performance at the end of a 4-year degree program. For each algorithm, experiments have been conducted using different combinations of the collected data.

  - The first set of experiments has been conducted using all the academic attributes provided in the Appendix.
  - The second set of experiments uses a combination of all the academic and derived attributes.
  - The demographic attribute of gender has been added to the existing attributes for the third set of experiments.

- The fourth set of experiments has been conducted using the feature selected subset of academic attributes (see Table 4).
- The fifth set of experiments has been conducted using the feature selected subset of academic and derived attributes.
- The sixth set of experiments has been conducted using the feature selected subset of academic, derived, and demographic attributes.

- For the discovery of an optimal classification model, the generated models have been evaluated using the metrices of Accuracy, F-Score (weighted average), and Kappa. The statistical difference in classifier performance has also been examined by means of $p$-value, computed using the Friedman test ($k$-1 degrees of freedom), to establish the significance of the results (*Settouti, El Amine Bechar & Amine Chikh, 2016*).
- After the selection of the preferred classification model, it is imperative to further explore the model to understand how the attributes influence the final prediction of a class label.

- To identify the courses that most significantly influence the academic performance of a student, the generated model has been visualized and examined in detail to understand the exact combination of the attributes and their precise configurations that lead to a particular class label.

- Finally, to identify students for intervention and necessary pedagogical actions, a segmentation framework in the form of a cross-tabular matrix has been proposed in this research.

## EXPERIMENTAL RESULTS AND DISCUSSION

As discussed in 'Feature Selection', the removal of attributes that inversely affect the prediction of the class label can greatly improve the quality of the generated model. CfsSubsetEval is a correlation-based feature evaluator used to find attributes that positively influence the prediction of the class label (*Hall, 1998*). As also explained, this study uses a total of 34 attributes (31 academic, two derived, and one demographic). Feature selection using CfsSubsetEval has been applied on three combinations of these collected attributes. Table 4 presents the resulting feature selected attributes obtained using CfsSubsetEval.

Observing Table 4, after the application of the CfsSubsetEval on the academic attributes, 17 out of the 31 academic attributes have been identified as attributes positively influencing the final prediction of student performance while predicting four class labels. Similarly, 19 out of the 31 academic attributes have been identified as attributes positively influencing the final prediction of student performance while predicting two class labels. The resulting subset of attributes after the application of CfsSubsetEval on the combination of academic and derived, and academic, derived, and demographic attributes for the classification of 4 and two class labels can also be seen in Table 4.

As explained in 'Knowledge Discovery', experiments have been conducted on different combinations of the collected data using widely popular classification algorithms. Table 5 presents the results of the various conducted experiments. The attribute set used with

each of the classifiers has been presented in the first column. The *p*-value for the observed classifier performance has been computed to monitor the statistical significance of the results. The statistical difference (*p*-value) for classifier performance has been presented in Table 6.

Observing the results in Tables 5 and 6, several classifiers exhibited good performance. While predicting four class labels, the most optimal performance was exhibited by the model generated by the NB classifier with an accuracy of 84.87%, weighted average $F_1$ score of 0.848, and a Kappa score of 0.7942. These results were obtained using a feature selected subset of academic, derived, and demographic attributes. The model generated by the RF classifier achieved the second highest accuracy of 83.50%. The RF classifier achieved these results while working under two configurations: (i) a feature selected subset of academic and derived attributes, and (ii) a feature selected subset of academic, derived, and demographic attributes. The SMO classifier exhibited an accuracy of 82.13% on a feature selected subset of academic and derived attributes, while the J48 classifier achieved an accuracy of 81.44% on a feature selected subset of academic and derived attributes. Unlike the NB classifier, RF, SMO, and J48 showed better performance when working with a feature selected subset of academic and derived attributes. Interestingly, apart from the model generated by the NB classifier, demographics have not been featured in the optimal model generated by any other classifier.

Focusing on the classifier performance under the six attribute configurations, it can be observed that the accuracy, $F_1$ score, and Kappa score of all six classifiers improved when working with a combination of academic and derived attributes. The NB classifier had an accuracy of 78.01% while working with only academic attributes; the accuracy of the classifier improved to 80.06% when working with a combination of academic and derived attributes. For the J48 classifier, the accuracy improved from 68.72% to 73.19%. Also, a vast improvement in the accuracy, $F_1$ score, and Kappa score of the classifiers can be observed when working on a feature selected subset of academic and derived attributes. The accuracy of the NB classifier improved to 84.53% while the J48 classifier exhibited an accuracy of 81.44% on a feature selected subset of academic and derived attributes. The improvement in the Kappa score of the classifiers can also be observed across the various configurations. For the J48 classifier, the Kappa score improved from 0.5744 while working with academic attributes to 0.7480 while working with a feature selected subset of academic and derived attributes.

An interesting observation is that the models based on the decision tree and rule-based classifiers have exhibited better performance when working with a combination of the academic and derived attributes; the performance of these classifiers has decreased when working with the demographic attribute of gender.

For the prediction of two class labels, the model generated by the SMO classifier exhibited the highest accuracy of 93.13%, weighted average $F_1$ score of 0.935, and a Kappa score of 0.8694. This performance was exhibited while working with a combination of academic, derived, and demographic attributes. The NB classifier had the second highest accuracy of 92.78% while working with a feature selected subset of academic and derived attributes.

**Table 5 Performance evaluation.**

| Classifier | Experiment - 4 classes | | | Experiment - 2 classes | | |
|---|---|---|---|---|---|---|
| | Accuracy (%) | $F_1$ score | Kappa score | Accuracy (%) | $F_1$ score | Kappa score |
| NB$_{(AC)}$ | 78.01 | 0.776 | 0.7015 | 89.35 | 0.893 | 0.7869 |
| NB$_{(AC+DR)}$ | 80.06 | 0.800 | 0.7288 | 90.72 | 0.907 | 0.8144 |
| NB$_{(AC+DR+DM)}$ | 79.72 | 0.796 | 0.7243 | 90.38 | 0.904 | 0.8075 |
| NB$_{(FS\ AC)}$ | 83.50 | 0.834 | 0.7761 | 91.41 | 0.914 | 0.8282 |
| NB$_{(FS\ AC+DR)}$[*] | 84.53 | 0.844 | 0.7897 | 92.78 | 0.928 | 0.8557 |
| NB$_{(FS\ AC+DR+DM)}$[a*] | 84.87 | 0.848 | 0.7942 | 92.78 | 0.928 | 0.8557 |
| J48$_{(AC)}$ | 68.72 | 0.684 | 0.5744 | 89.00 | 0.890 | 0.7801 |
| J48$_{(AC+DR)}$ | 73.19 | 0.732 | 0.6333 | 89.00 | 0.890 | 0.7801 |
| J48$_{(AC+DR+DM)}$ | 72.16 | 0.722 | 0.6183 | 89.35 | 0.893 | 0.7870 |
| J48$_{(FS\ AC)}$ | 70.44 | 0.703 | 0.5958 | 89.00 | 0.890 | 0.7801 |
| J48$_{(FS\ AC+DR)}$[a*] | 81.44 | 0.814 | 0.7480 | 90.72 | 0.907 | 0.8145 |
| J48$_{(FS\ AC+DR+DM)}$[*] | 79.72 | 0.796 | 0.7244 | 90.72 | 0.907 | 0.8145 |
| JRip$_{(AC)}$ | 70.44 | 0.702 | 0.5973 | 87.97 | 0.880 | 0.7595 |
| JRip$_{(AC+DR)}$[*] | 77.66 | 0.773 | 0.6977 | 90.38 | 0.904 | 0.8075 |
| JRip$_{(AC+DR+DM)}$ | 72.16 | 0.719 | 0.6214 | 89.35 | 0.893 | 0.7870 |
| JRip$_{(FS\ AC)}$ | 71.13 | 0.707 | 0.6069 | 90.03 | 0.900 | 0.8007 |
| JRip$_{(FS\ AC+DR)}$[a] | 74.91 | 0.749 | 0.6561 | 88.32 | 0.883 | 0.7663 |
| JRip$_{(FS\ AC+DR+DM)}$ | 73.53 | 0.731 | 0.6396 | 86.94 | 0.869 | 0.7388 |
| RF$_{(AC)}$ | 81.09 | 0.812 | 0.7397 | 91.07 | 0.911 | 0.8213 |
| RF$_{(AC+DR)}$[a] | 83.50 | 0.835 | 0.7737 | 90.72 | 0.907 | 0.8144 |
| RF$_{(AC+DR+DM)}$ | 80.75 | 0.808 | 0.7360 | 91.07 | 0.911 | 0.8213 |
| RF$_{(FS\ AC)}$[*] | 71.13 | 0.707 | 0.6069 | 90.03 | 0.900 | 0.8007 |
| RF$_{(FS\ AC+DR)}$[a] | 83.50 | 0.836 | 0.7731 | 91.07 | 0.911 | 0.8213 |
| RF$_{(FS\ AC+DR+DM)}$[a] | 83.50 | 0.835 | 0.7731 | 90.72 | 0.907 | 0.8144 |
| SMO$_{(AC)}$ | 79.72 | 0.798 | 0.7211 | 92.09 | 0.921 | 0.8419 |
| SMO$_{(AC+DR)}$ | 81.78 | 0.819 | 0.7494 | 92.44 | 0.924 | 0.8488 |
| SMO$_{(AC+DR+DM)}$[*] | 81.78 | 0.818 | 0.7498 | 93.47 | 0.935 | 0.8694 |
| SMO$_{(FS\ AC)}$ | 81.44 | 0.816 | 0.7443 | 91.75 | 0.918 | 0.8351 |
| SMO$_{(FS\ AC+DR)}$[a] | 82.13 | 0.822 | 0.7537 | 92.78 | 0.928 | 0.8557 |
| SMO$_{(FS\ AC+DR+DM)}$ | 81.78 | 0.819 | 0.7496 | 93.13 | 0.931 | 0.8625 |
| KStar$_{(AC)}$ | 72.51 | 0.726 | 0.6227 | 87.29 | 0.873 | 0.7457 |
| KStar$_{(AC+DR)}$ | 75.94 | 0.760 | 0.6700 | 88.32 | 0.883 | 0.7663 |
| KStar$_{(AC+DR+DM)}$ | 76.28 | 0.764 | 0.6745 | 87.97 | 0.880 | 0.7594 |
| KStar$_{(FS\ AC)}$[a*] | 76.97 | 0.771 | 0.6823 | 89.35 | 0.893 | 0.7869 |
| KStar$_{(FS\ AC+DR)}$ | 75.60 | 0.757 | 0.6656 | 87.29 | 0.873 | 0.7457 |
| KStar$_{(FS\ AC+DR+DM)}$ | 75.26 | 0.754 | 0.6605 | 89.00 | 0.890 | 0.7800 |

**Notes.**

[a] indicates the result with highest accuracy generated by a classifier when predicting four.

class labels and [*] indicates the result with the highest accuracy generated by a classifier when predicting two class labels.

**Table 6  Statistical difference in classifier performance.**

| Experiment - 4 classes | | Experiment - 2 classes | |
|---|---|---|---|
| Classifier | *p*-value | Classifier | *p*-value |
| NB vs J48* | 0.01431 | NB vs J48* | 0.01431 |
| NB vs JRip* | 0.01431 | NB vs JRip* | 0.01431 |
| NB vs RF | 1 | NB vs RF | 0.68309 |
| NB vs SMO | 1 | NB vs SMO* | 0.04123 |
| NB vs KStar* | 0.01431 | NB vs KStar* | 0.01431 |
| J48 vs JRip | 0.68309 | J48 vs JRip | 0.68309 |
| J48 vs RF* | 0.01431 | J48 vs RF* | 0.04123 |
| J48 vs SMO* | 0.01431 | J48 vs SMO* | 0.01431 |
| J48 vs KStar | 1 | J48 vs KStar | 0.10247 |
| RF vs JRip* | 0.04123 | RF vs JRip* | 0.04123 |
| RF vs KStar | 0.10247 | RF vs KStar* | 0.01431 |
| JRip vs Kstar | 0.10247 | JRip vs Kstar | 0.10247 |
| SMO vs JRip* | 0.01431 | SMO vs JRip* | 0.01431 |
| SMO vs RF | 0.41422 | SMO vs RF* | 0.01431 |
| SMO vs KStar* | 0.01431 | SMO vs KStar* | 0.01431 |

**Notes.**

*$p$-value significant $p \leq 0.05$.

Looking at the results presented in Table 5, it can be seen that the accuracy, $F_1$ score and Kappa scores have greatly improved when working with two class labels. The accuracy of models generated by all six classifiers is approximately equal to or above 90%. The Kappa scores have also greatly improved. Similar to the experiments of predicting 4 class labels, the model with the highest accuracy while predicting two class labels has been built using academic, derived, and demographic attributes. Most classifiers have shown an improvement when working with a feature-selected subset of attributes.

The results presented in Table 5 demonstrate that it is possible to generate a model for the early detection of student end-of-degree performance using the most basic and readily available learning data collected by higher educational institutes. Thus, the first research question has been answered in the affirmative.

### Classification model

Even though experiments have been conducted with several classifiers, as previously established by *Asif et al. (2017)*, and discussed in 'Classification', the target courses under their exact configurations cannot be identified with all classifiers. Keeping in mind that a goal of this research is not only the early prediction of student academic performance, but also early intervention through the identification of courses that play a significant role in influencing the academic performance of a student, a trade-off needs to be made between classifier accuracy in favour of the interpretability of the model.

The results of the decision-tree classifier J48 have been considered here to identify courses that can help educators provide the necessary intervention, at an early stage, to at-risk students. Due to the extensive size of the generated model (tree) for classification of

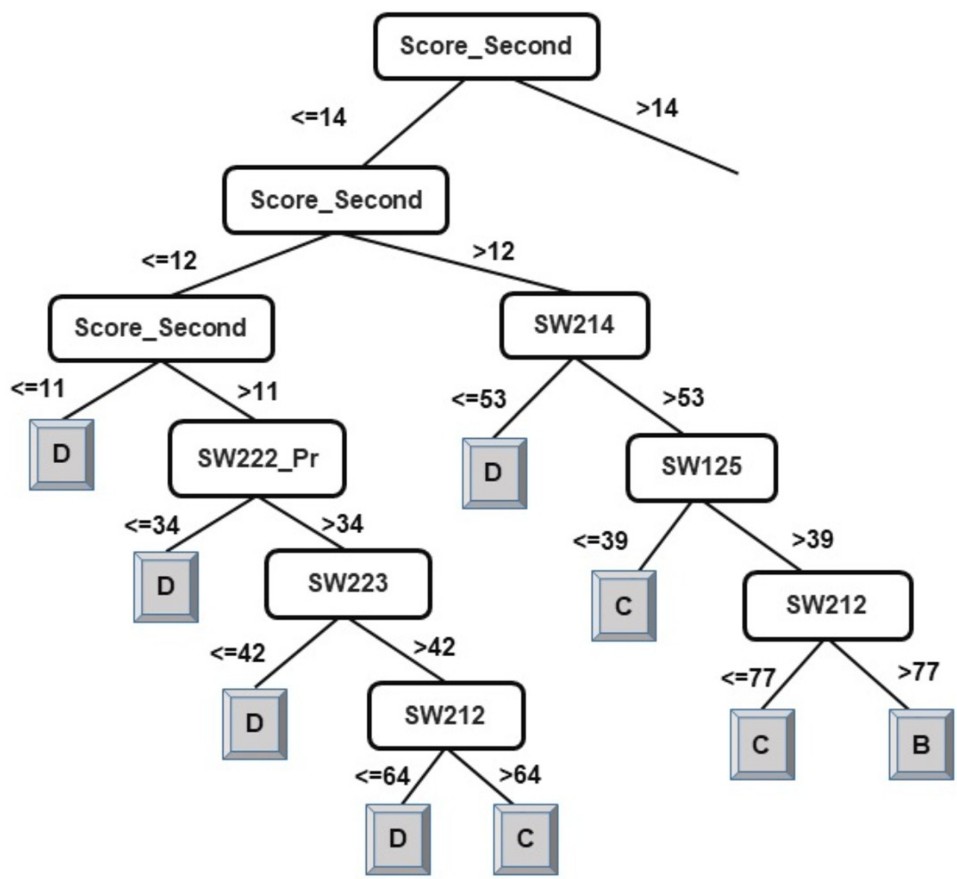

**Figure 2  J48 model for predicting the class of learners at the end-of-degree (I).**

students into four classes, it has been split in two parts. The left-side of the J48 tree for the classification of students into four classes has been presented in Fig. 2 and the right-side of the J48 tree has been presented in Fig. 3.

As explained in the 'Classification' section, the root node of a decision tree identifies the attribute which most strongly influences the final prediction of the class label. Similarly, nodes at a higher level in the tree (closer to the root node) play a stronger role in influencing the final class label. From the model in Figs. 2 and 3, it can be observed that the derived attribute of the accumulated score at the end of the 2nd year is the most important feature towards the final prediction of student performance. Several paths can be taken along the tree to reach the leaf nodes. The most direct paths leading to the leaf nodes D, C, and B have been presented in Fig. 4.

Following the blue arrows in Fig. 4A), if a student has a 2nd year accumulated score of greater than 12 and scores less than 53 marks in the course SW214, the resulting class of the student will be D. Similarly, following the arrows in Fig. 4B), if a student has a 2nd year accumulated score of greater than 12, more than 53 marks in the course SW214 and a score of 39 marks in the course SW125, the resulting class of the student will be C. The

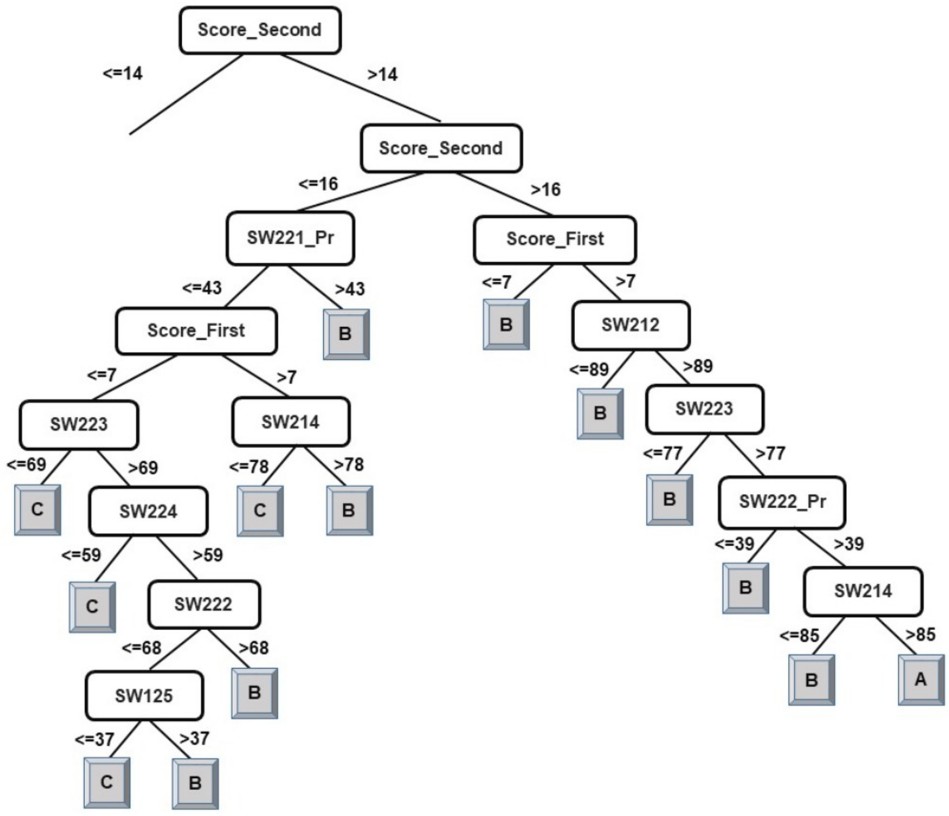

**Figure 3  J48 model for predicting the class of learners at the end-of-degree (II).**

**Figure 4  Courses influencing the final prediction of student end-of-degree performance.**

performance of a student in these courses directly influences the final performance class of a student.

Thus, observing Figs. 2 and 3, the courses SW214, SW125, SW221_Pr, SW222, SW212, and SW223 have been identified as the main courses that affect student end-of-degree performance while classifying students into four classes.

Following the same logic, when observing the J48 tree for the classification of students into two classes (see Fig. 5), the derived attribute of the accumulated score at the end of the

2nd year is the most important feature towards the final prediction of student performance into two classes. The courses SW125, SW221_Pr, EL101_Pr, SW214, SW223, and SW211 have been identified as playing a key role in the final prediction of student end-of-degree performance. Some interpretations that can be made from the model presented in Fig. 5 are:

- Having a 2nd year accumulated score of less than or equal to 14 and a score of less than or equal to 39 in SW125 will result in graduating under Class NI.
- Having a 2nd year accumulated score of between 14–15 and a score of greater than 43 in EL101_Pr will result in graduating under Class SP.

An examination of the model presented in Figs. 2, 3, and 5 answers research question two. It is now safe to conclude that courses strongly influencing the final prediction of student end-of-degree performance can be ascertained.

### *Segmentation framework*

To identify students for intervention and necessary pedagogical actions, a segmentation framework in the form of a cross-tabular matrix has been proposed. To generate the segmentation matrix, student academic performance at the end of the 2nd year of their university education has been computed. Using the classification model, student performance at the end of the degree has been predicted. The segmentation matrix confronts the observed student performance at the end of the 2nd year against the final performance predicted by the model. As this research uses two approaches to classify students, two segmentation matrixes have been generated. Figure 6 presents the segmentation matrix where students have been classified into two classes based on the percentage obtained at the end of the degree: SP (satisfactory performance: $> = 75\%$) or NI (needs improvement: $< = 74\%$).

Evident from Fig. 6, a majority of students stay in the same segment at the end of the degree as they did at the end of the 2nd year of the degree program: 121 students reside in the satisfactory performance segment, and 136 students reside in the needs improvement segment. 16 students have moved from the satisfactory segment to the needs improvement segment. The segmentation matrix raises two main concerns. First, a very large proportion of students (136) is persistently performing below a satisfactory performance level. Second, 16 students that resided in the satisfactory segment up until their 2nd year fall into the needs improvement segment by the completion of their degree. As evident from their prior results, these students have the potential to perform better. The students in these two segments are being neglected by the educational institute. A system of feedback, intervention, mediation, and active involvement of the instructors and policy-makers can help students move from these segments.

Using two classes allows us to understand student performance to a small extent. However, bifurcating these classes into further subdivisions will help pinpoint students across various performance levels. Figure 7 presents the segmentation matrix where students have been classified using the second approach. Here, students have been segregated into

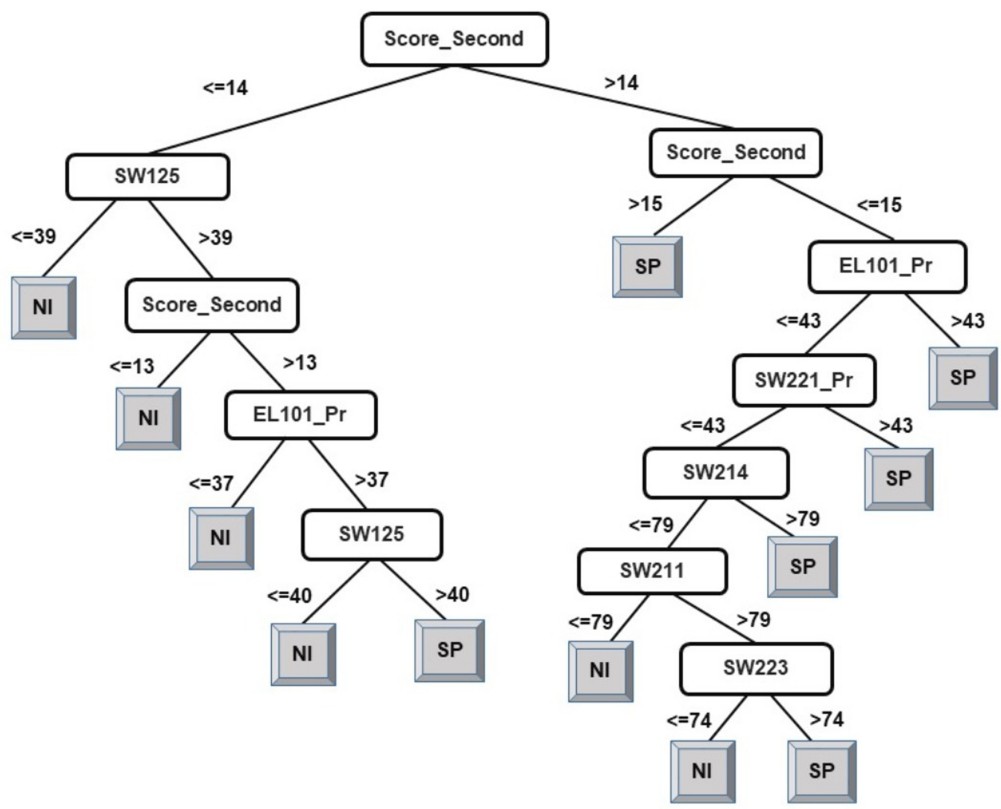

**Figure 5** J48 model for predicting the class of learners at the end-of-degree (two classes).

four classes: A (high achievers: $>= 85\%$), B (above-average achievers: 75%–84%), C (average achievers: 65%–74%), and D (underachievers: $<65\%$).

Observing the diagonal of the segmentation matrix in Fig. 7, most students graduate in the same performance segment they belonged to at the end of their 2nd year: 43 students in Class-A, 63 in Class-B, 50 in Class-C, and 55 in Class-D. The cells adjacent to the diagonal identify students whose performance changes after the 2nd year. Observing the last row of the segmentation matrix, 13 students that were in Class-A at the end of the 2nd year are predicted to finish their education in Class-B, and 4 students that reside in Class-A are predicted to finish their education under Class-C. Observing the second row from the top, 21 students who reside in Class-C at the end of the 2nd year have been predicted to complete the degree in the Class-B performance segment. These students have potential, and perhaps having the right pedagogical strategies may help them jump up to the high-achiever segment. A major concern in the segmentation matrix is the top-right cell: 55 students that are predicted to complete their degree as underachievers in Class-D.

The suggested approach identifies 16 student segments allowing the institute to design a pedagogical policy to specifically target each segment. A robust, pragmatic policy can be devised to mitigate factors that lead to poor performance levels and identify academically motivated students. Using the approach proposed in this research, it can be concluded that

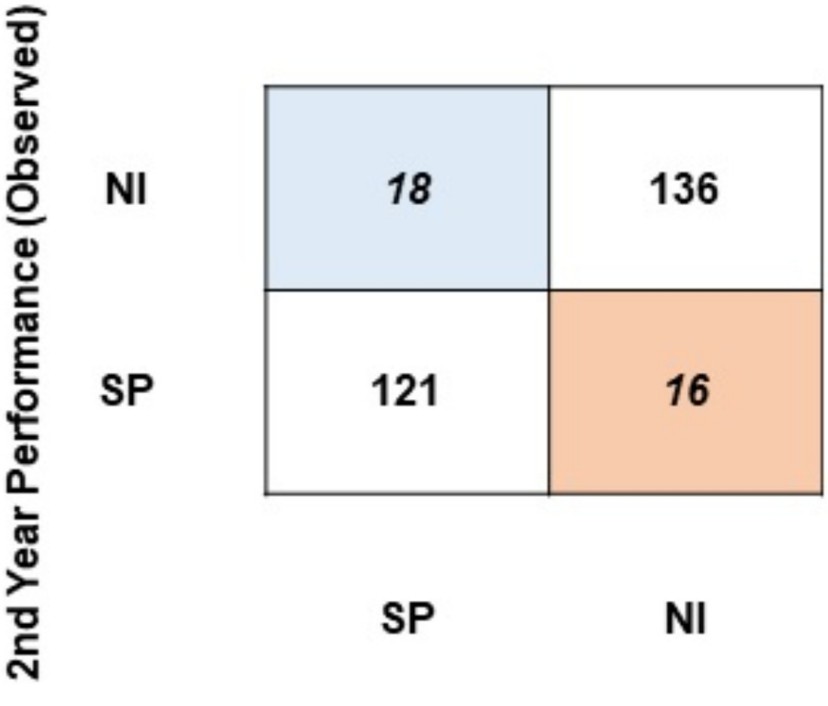

**Student Segmentation Matrix**

**Figure 6** **Student segmentation matrix (two classes).**

a segmentation framework based on student performance can be devised to help design a pragmatic pedagogical policy.

## Discussion

Higher educational institutes collect and store multiple facets of student data. The analysis of this data has the potential of uncovering patterns and insights that can shape pedagogical policies. The aim of this research has been the exploration of the most basic data collected by higher educational institutes to devise a classification model that can predict student end-of-degree performance at an earlier stage during the course of the degree. Consistent with the research conducted by *Nghe, Janecek & Haddawy (2007)*, *Asif et al. (2017)*, *Miguéis et al. (2018)*, *Nieto, García-Díaz & Montenegro (2019)* and *Aman et al. (2019)*, the current research validates that it is possible to successfully predict student performance at the end of a degree program using student data at some earlier point during the course of the degree.

As the reviewed studies differed in the attributes used, efforts were made in the current research to conduct experiments that would build upon concepts provided in

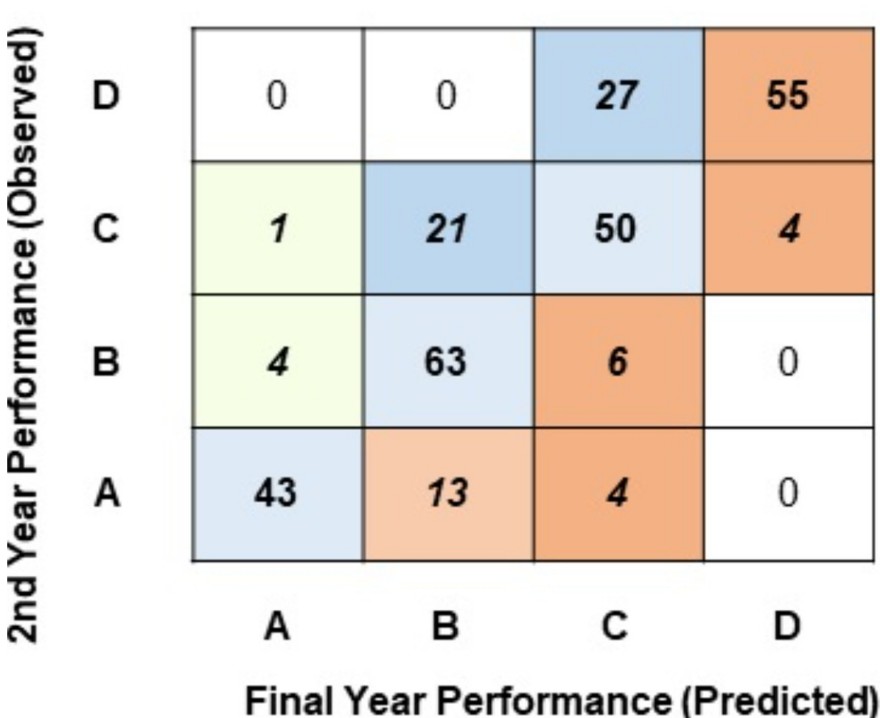

**Student Segmentation Matrix**

**Figure 7** **Student segmentation matrix (four classes).**

the aforementioned studies. To this end, the first set of experiments focused only on using academic attributes (marks in SSC, HSC, university admission test, and the marks in subjects studied in the first two years of the degree program). The results outlined in Table 5 clearly indicate that consistent with the studies conducted by *Miguéis et al. (2018)* and *Asif et al. (2017)*, student performance can be predicted using only academic attributes. However, it needs to be noted that experimenting with a feature-selected subset of academic attributes greatly improved the performance of the classifiers. In the case of the NB classifier, the performance in terms of accuracy increased from 78.01% to 83.50% when predicting 4 classes and from 89.35% to 91.41% when predicting two classes. Similarly, the Kappa score of the NB classifier improved from 0.7015 to 0.7761 when predicting 4 class labels and from 0.7869 to 0.8282 when predicting 2 class labels.

The current research computed two new attributes of accumulated scores at the end of the 1st and 2nd year of the degree program. The addition of these derived attributes significantly improved the classifiers' performance. While predicting four class labels, the J48 classifier exhibited an accuracy of 68.72% using only academic attributes, which improved to 73.19% with the addition of the derived attributes. This accuracy further increased to 81.44% when a feature-selected subset of academic and derived attributes

was used. Similarly, for the NB classifier, an accuracy of 84.53% was observed on a feature-selected subset of academic and derived attributes.

The addition of the attribute of gender did not play a significant role in the final prediction for all classifiers. Although the addition of this attribute improved the performance of the NB and SMO classifiers, it had the opposite effect on the performance of the J48, KStar, and JRip classifiers. Thus the conflict of using the attribute of gender still stands (*Khan & Ghosh, 2021*). Consistent with the findings of *Asif et al. (2017)*, *Zimmermann et al. (2015)*, and *Nghe, Janecek & Haddawy (2007)*, it was also observed that classifier performance is inversely proportional to the number of class labels. The lesser the number of class labels, the better the performance of the classifier; all classifiers in the current study exhibited better results when classifying student performance into two class labels. Also, consistent with the conclusion established while reviewing the previous studies, there is no 'best' classification algorithm for the prediction of student performance. The nature of the data being explored and the number of class labels being predicted greatly influence the performance of a classifier.

At the end of the experiments, it can be concluded that a classification model to predict the class a student will graduate in can successfully be generated with a subset of academic and derived attributes. Using feature selection greatly improves the classifiers' overall performance and can aid in reducing the complexity of the final model.

The entire purpose of predicting student end-of-degree performance at an earlier stage during the course of the degree program is to ensure that students can be provided help and intervention to ensure that they maximize their full potential. As pointed out by *Xiao, Ji & Hu (2022)*, the interpretation of the generated model is an important step towards providing feedback for academic improvement. Visualizing the generated model allows an understanding of how student performance in various courses influences the final academic class of a student. An interpretation of the J48 decision tree has been presented in this article to identify courses that strongly influence the final performance of a student. Instructors can monitor student performance in the highlighted courses and provide the needed feedback for early intervention.

In order to identify students for intervention and necessary pedagogical actions, a segmentation framework in the form of a cross-tabular matrix has also been proposed in this research. The goal here was to confront the observed student performance at the end of the 2nd year of the degree program against the final performance predicted by the model. The suggested approach in this article identifies 16 student segments and allows instructors to foresee the academic trajectory of each student. The use of the proposed framework will provide instructors and policy makers a mechanism of identifying students at various levels of academic performance along with the knowledge of how this performance will fluctuate over the course of the degree. A robust, pragmatic policy can, thus, be devised to mitigate factors that lead to poor performance levels and even identify academically motivated students. A proactive intervention policy that specifically targets each performance segment can also be devised.

# CONCLUSION AND FUTURE WORK

This research explores and analyzes the most basic student data available in a 4-year degree program. Three research questions have been investigated in this article. The first question focused on the generation of a classification model for early identification of student end-of-degree performance using the most basic and readily available learning data collected by higher educational institutes. It was observed that student performance at the end of a degree program could successfully be predicted using a feature-selected combination of academic and derived attributes. The second question focused on deriving courses that strongly influence the final prediction of student performance. The model generated using the J48 classifier has been used to indicate the courses that influence the final prediction of student performance. Furthermore, the marks obtained in these courses can be used to classify students into various performance levels and thus be used to provide intervention to students at risk of obtaining poor grades. The third question involved the generation of a segmentation framework. A cross-tabular segmentation matrix has been used to confront the computed student performance at the end of the 2nd year against the final performance as predicted by the generated model. The resultant segmentation matrix identifies students in various performance segments. The early identification of these students provides the opportunity to robustly devise a pragmatic policy to specifically target each performance level.

This research aims to provide instructors and policymakers with the much-needed feedback to truly create a student-centric learning environment. Several courses have been identified as indicators of student performance in this research. An important future direction can be to explore student performance in these courses. This will provide the educational institute an added opportunity to improve educational outcomes. Also, using the approach outlined in this article, predictive models can be built for the early identification of student performance across the other degree programs offered by the university. The early prediction of student performance will help in designing a pedagogical policy that can increase the quality of education by not only mitigating academic failure but also by encouraging higher performance.

# APPENDIX

| Attribute | Description | Value |
| --- | --- | --- |
| Gender | Student gender | M-F |
| SSC | SSC Exam Marks | 0–850 |
| HSC | HSC Exam Marks | 0–1100 |
| Ad_Test | University Admission Test Marks | 0–100 |
| ENG11 | Functional English | 0–100 |
| MTH108 | Applied Calculus | 0–100 |
| SW111 | Computer Programming | 0–100 |
| SW111_Pr | Computer Programming Practical | 0–50 |

| Attribute | Description | Value |
|---|---|---|
| ES121 | Electronics Engineering | 0–100 |
| ES121_Pr | Electronics Engineering Practical | 0–50 |
| ES101 | Electrical Engineering | 0–100 |
| ES101_Pr | Electrical Engineering Practical | 0–50 |
| MTH112 | Linear Algebra & Analytical Geometry | 0–100 |
| SW121 | Data Structures & Algorithms | 0–100 |
| SW121_Pr | Data Structures & Algorithms Practical | 0–50 |
| SW122 | Digital Computers & Logic Design | 0–100 |
| SW122_Pr | Digital Computers & Logic Design Practical | 0–50 |
| SW125 | Professional Ethics | 0–50 |
| MTH212 | Differential Equations & Fourier Series | 0–100 |
| SW214 | Information Systems | 0–100 |
| SW215 | Software Economics & Management | 0–50 |
| SW224 | Computer Architecture & Organization | 0–100 |
| SW211 | Operating Systems Concepts | 0–100 |
| SW211_Pr | Operating Systems Concepts Practical | 0–50 |
| MTH217 | Laplace Transform & Discrete Mathematics | 0–100 |
| SW223 | Operations Research | 0–100 |
| SW221 | Object Oriented Programming | 0–100 |
| SW221_Pr | Object Oriented Programming Practical | 0–50 |
| SW212 | Microprocessor Technology | 0–100 |
| SW212_Pr | Microprocessor Technology Practical | 0–50 |
| SW222 | Database Management Systems | 0–100 |
| SW222_Pr | Database Management Systems Practical | 0–50 |
| Score_First | 1st Year Accumulated Score | 0–10 |
| Score_Second | 2nd Year Accumulated Score | 0–20 |

### Funding

The authors received funding from the Deanship of Scientific Research at Najran University for this research through a grant (NU/RG/SERC/12/23) under the Research Groups Funding program at Najran University, Kingdom of Saudi Arabia. The funders had no role in study design, data collection and analysis, decision to publish, or preparation of the manuscript.

### Grant Disclosures

The following grant information was disclosed by the authors:
The Deanship of Scientific Research at Najran University for this research under the Research Groups Funding program at Najran University, Kingdom of Saudi Arabia: NU/RG/SERC/12/23.

### Competing Interests

The authors declare there are no competing interests.

## Author Contributions

- Areej Fatemah Meghji conceived and designed the experiments, analyzed the data, authored or reviewed drafts of the article, and approved the final draft.
- Naeem Ahmed Mahoto conceived and designed the experiments, analyzed the data, authored or reviewed drafts of the article, and approved the final draft.
- Yousef Asiri conceived and designed the experiments, analyzed the data, authored or reviewed drafts of the article, and approved the final draft.
- Hani Alshahrani conceived and designed the experiments, performed the experiments, analyzed the data, performed the computation work, prepared figures and/or tables, and approved the final draft.
- Adel Sulaiman performed the experiments, performed the computation work, prepared figures and/or tables, and approved the final draft.
- Asadullah Shaikh performed the experiments, performed the computation work, prepared figures and/or tables, and approved the final draft.

## Ethics

The following information was supplied relating to ethical approvals (i.e., approving body and any reference numbers):

Advanced Studies and Research Board (ASRB), in its 136th meeting at Mehran University of Engineering and Technology, approved this study with reference resolution number 136.43

## Data Availability

The raw data is available in the Supplemental File.

## Supplemental Information

Supplemental information for this article can be found online at http://dx.doi.org/10.7717/peerj-cs.1294#supplemental-information.

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
