# Peer review of "Early detection of student degree-level academic performance using educational data mining"

_PeerJ Computer Science, doi:10.7717/peerj-cs.1294_

## Round 0.1 · original submission · Major Revisions

Comments from Editor:

Please clarify the third contribution regarding the early detection of the student academic performance in terms of the influenced courses.

I suggest the authors to draw a figure clarifying the authors' model.

Reviewer 1 ·

Basic reporting

This study has merit, but there are two important issues I like to raise with the editorial team before I can recommend acceptance with revision:
1) WARNING ETHICS: this is using human derived data, there must be ethics approval for such a study. Publication should not proceed until the question of ethical approval to access the database and use data has been clearly stated. The data refers to individual results and as this study seeks to develop a predictor of performance for “a student”, ethics is an absolute necessity.
2) Data availability maybe no in line with the journal requirements. The authors state table 1, 2, 4 hold the data, but I can see no raw data in table 1., basic student numbers in table 2, and the results of the analysis in table 4. This does not constitute the data needed to replicate the study
One these issues are clarified I CANNOT recommend acceptance without significant (MAJOR) revisions outline below, particularly in relation to currency of literature, style errors, lack of statistical supports, and some areas that are not clear as to their meaning or context to the paper these have been noted in the general comments below.

Experimental design

General comments
This paper uses classifiers to predict the performance of students at university based on their prior grades. They demonstrate that this is possible.
One significant problem with the article is the lack of current references. The currency of the literature (age of the articles), only one is 2021, the majority much earlier. The authors need to update the literature in this paper reflect the possible changes in the theory that surround this topic.
There are significant formatting issues with the intext references throughout Eg Bransford et al. 1999 should be (Bransford et al. 1999). All these need to be corrected.
What is the university policy (ln 261) and how does that affect the data. The paragraph (ln. 261-264) is totally confusing. I have no idea what the 10% of 1st year means, or how that related to the study this needs to explained much clearer.
(ln 326) why cant this be derived? How does this affect the ability to specifically target students. What are the trade-offs mean for the practical application.
The references need to be formatted correctly for the journal. There are inconstancies with the style (eg. ln 542)
This paper uses linear modelling (p. 185) in which case there needs to be correlation statistics presented. And these need to be tested (r2) etc.
The presentation of F statistics is not sound these need to be presented correctly, that is we need the p-value for each, identification of the Degrees of Freedom, N-DOF.
(ln 280) what does “ CfsSubsetEval “ mean
(in 298-299) this needs to be contained in the table caption “Please note that α indicates the best result generated by a classifier when predicting four class labels, and ∗ indicates 299 the best result generated by a classifier when predicting two class labels” and should be reworded, and Further. the table heading “experimental results” is meaningless.

Validity of the findings

The authors state “The NB has the best result” but there is no test of the statistical difference between the set of data, and therefore this claim is subjective.

Additional comments

.

Annotated reviews are not available for download in order to protect the identity of reviewers who chose to remain anonymous.

·

Basic reporting

well-addressed, excepted the Literature references format. Please see the attached document (please make it available for the author) for details.

Experimental design

Secondary research questions are well defined, and are encompassed by the research question clearly announced in those abstract and introduction . Rigorous investigation is performed (see the attached document for more details). Methods are well described with sufficient detail and information to replicate (see the attached ocument)

Validity of the findings

addressed (see details on the document attached)

Reviewer 3 ·

Basic reporting

1.The methodology of the research is not clearly stated. Your research paper could be improved by adding this section and describing how you conducted your research. Additionally, they must also elaborate on the methods and procedures used to collect, explore, analyze and present the data. Both of these things should be stated in detail.
2. This research paper needs to be improved in terms of the English language.
3. Literature reviews should be improved and written in a different format. The purpose of this is to ensure that the ideas in the research are linked so that the reader can easily identify the gaps and that will improve the efficiency of the section.
4.It would be best if the objectives and research hypotheses were put in a separate section of the research paper so that they are obvious to the reader.
5.It is necessary to revise the way in which references are referred to
6.It is necessary to place table 3 after the paragraph on page 11

Experimental design

1. Neither the source of the data nor the structure of the data are mentioned.
2. Neither the source of the data nor the structure of the data are mentioned. In order to convey information to the reader about the characteristics of the data set, I recommend that the author revise the section on data collection. It would be beneficial if this section contained a table that provided more detailed information regarding the characteristics. Additionally, it will be easier for the reader to understand what will be discussed in the section on features selection.
3. It is necessary to explain the procedure and methods of integrating the data in detail
4.It is not stated whether the data set has missing entries, or what will happen if the data is missing. The procedure and methods for handling missing data should be added. It is also recommended to include a table that shows statistically how many missing data are available for each column.

Validity of the findings

1. In the classification model selection, there is no indication as to why the J48 model has been selected as the best model .
2. A more detailed discussion of the findings is required

Additional comments

It is not stated explicitly how the research was conducted. It would be a good idea to include this section and outline the research methodology in the research paper. The data must also be presented in a detailed manner, outlining what techniques were used and how the data was collected, investigated, evaluated, and presented. In order to fully understand all these points, the authors need to go into detail about them.

---

## Round 0.2 · Minor Revisions

1. A more detailed discussion of the findings is required
2. Table 4 and Table 5 should be discussed further.
3. Further discussion of knowlege discovery should be discussed more. I suggest mentioning this discussion as points style.
4. Also you didn't reply to my comment below clearly:
Please clarify the third contribution regarding the early detection of the student academic performance in terms of the influenced courses. I suggest the authors to draw a figure clarifying the third contribution in their research.

---

## Round 0.3 · accepted · Accept

I confirm that the authors have addressed all of the reviewers' comments and this manuscript is ready for publication.